# Linear Recurrent Neural Networks with a Feature-Sequence Twist

## Abstract

The transformer network architecture has led to advances in artificial intelligence. Conversational AI applications, such as ChatGPT, and protein folding predictions with AlphaFold are made possible by transformer architectures and the self-attention mechanism. However, advancing towards more general, flexible, and energy-efficient artificial intelligence may require exploring new architectures that differ significantly from those currently used. Transformer networks have largely replaced recurrent neural networks (RNNs) for state-of-the-art performance on sequence-based tasks. However, in recent years there has been some successful competition from linear recurrent neural networks (LRNNs) and state space models (SSMs). A core advantage of LRNNs and SSMs over traditional RNNs is that the hidden states can be calculated in parallel. Therefore, like the transformer, they can make efficient use of GPU computation. Unlike the transformer, computational costs of parallelized LRNNs and SSMs can scale sub-quadratically with sequence length. Despite these advantages, LRNNs and SSMs often struggle to generate the deep and rich representations that have contributed to the success of transformer architectures. We introduce Feature-Sequence Twisting (FST), a novel technique that transposes the sequence and feature dimensions between LRNN blocks. The purpose of FST is to generate deeper representations of the sequence in subsequent LRNN blocks. Since the computational cost of LRNNs scale sub-quadratically with sequence length, FST remains practical to compute even for large feature dimensions. Our experiments demonstrate that the FST architecture outperforms transformer networks on tasks such as Long ListOps, achieving performance competitive with state-of-the-art models.

## 1 Introduction

Advancements in artificial intelligence may require exploring architectures that differ significantly from those currently used. However, the search for novel architectures is challenging due to the vast range of network architectures that could be explored. Biological neural networks comprising the human brain and nervous systems of other complex species are capable of cognitive flexibility, rapid learning ability, and high energy efficiency. We can use known biological structures as a guide to navigate the possibilities for artificial neural networks, as argued in Hassabis et al. (2017).

Recurrent neural networks (RNNs) are an important class of artificial neural networks inspired by the recurrent connectivity observed in biological neural systems, such as those in the mammalian neocortex. This recurrence, and the neural dynamics which arise from it, are central to diverse computations and functions in the brain, including short-term memory, attention, and response normalization. RNNs are thus important tools for modeling neural circuit functions in computational neuroscience studies, and have many applications in AI, particularly for solving tasks that require integrating or comparing information across time. The basic RNN works by updating the activity of a hidden state $h_t$ at time-step $t$ with the equation

$$h_t = f(W_{rnn}h_{t-1} + W_{in}x_t + b),$$

where $f$ is the activation function, $W_{rnn}$ is the recurrent matrix, $W_{in}$ is the input matrix, $x_t$ is the input at time-step $t$ and $b$ is the bias. These networks are designed for handling sequential inputs $x_{0,1,2,...,T}$. The activation function in RNNs can be loosely compared to the nonlinear response

of biological neurons to synaptic inputs, with hidden state activities representing the firing rates of neurons or neural populations. Early examples of recurrent networks include Hopfield networks introduced in Hopfield (1982), which were used to model associative memory. Later, the ability of artificial RNNs to perform well at sequence-based tasks expanded further with the introduction of more advanced architectures using multiple hidden states, such as long short-term memory networks (LSTMs) introduced in Hochreiter & Schmidhuber (1996).

Introduced in the paper 'Attention is All You Need', Vaswani et al. (2017), the transformer network has revolutionized AI for sequence based tasks. Transformer architectures are the underlying networks behind products like ChatGPT, AlphaFold, and more effective versions of Google Translate. The key innovation in this architecture is the "self attention" (SA) mechanism. This works by calculating 'attention weights' which transmit information about how elements of a sequence relate to one another. It is clear that the transformer is not directly analogous to biological neural networks; notably, the transformer network does not involve recurrence, which is thought to be crucial to the way the brain processes sequential information.

Contrary to the argument that we need new architectures inspired by biological neural networks, much progress has been made by simply scaling up the size of transformer networks and the data used to train them. Much of the transformer's success comes from the computational efficiency of its large matrix multiplication operations, which are well-suited for GPU-based training. The transformer is also good at providing deep representations of sequences by capturing long-range dependencies. However, transformers do have some important limitations. One such limitation is that the SA mechanism does not inherently capture sequence order and requires positional encoding to provide positional information for each element. When using RNNs, we naturally preserve information about sequence order in the dynamics of recurrent network activity. The parallel structure of the transformer is also a double-edged sword, as the operations scale with the square of the sequence length. We aim to construct RNNs that offer the same parallel GPU training efficiency, provide deep representations of sequences, and scale sub-quadratically with sequence length. By doing so, we hope to maintain the advantages of transformers without some of their limitations.

We will describe some of the advantages of using a linear recurrent neural network (LRNN). These networks are simple and well-studied, but they have several unique properties that make them highly relevant for comparison with transformers. By leveraging LRNNs as building blocks and applying several computational and structural tricks, we demonstrate how this network type can outperform transformer networks.

Traditionally, when using RNNs, the representation of the sequence dimension is not changed by treating it as a set of features. What sets this work apart from other approaches is that we aim for subsequent layers of the network to contain deeper representations of the sequence dimension. We introduce a novel Feature-Sequence Twisting (FST) layer. In this layer, we transpose the sequence and feature dimensions between LRNN blocks. We find that networks using the FST blocks can perform well on the WikiText-103 language task and the Long ListOps task, designed to test a model's ability to handle long-range dependencies.

## 2 NETWORK STRUCTURE

### 2.1 LINEAR RECURRENT NEURAL NETWORKS

As before, we can describe a simple RNN by

$$h_t = f(W_{rnn}h_{t-1} + W_{in}x_t + b).$$

If we use a linear activation function and set the bias to zero we get

$$h_t = W_{rnn}h_{t-1} + W_{in}x_t.$$

If we substitute $h_{t-1}$ for the activity at $t-2$ we get

$$h_t = W_{rnn}^2 h_{t-2} + W_{rnn}W_{in}x_{t-1} + W_{in}x_t.$$

By continuing this recursively until we get to the initial activity $h_0$ we get the relationship

$$h_t = W_{rnn}^t h_0 + \sum_{i=0}^{t-1} W_{rnn}^i W_{in}x_{t-i}.$$

We can use this equation to take a series of inputs $x_{0,1,2...}$ and calculate the set of activities $h_{1,2,3...}$ in parallel, without the need to compute intermediate states. This property has been long known, but it is one of the reasons that linear RNNs have been gaining more attention recently in studies such as Orvieto et al. (2023).

The parallel calculation is made much simpler and equally expressive if $W_{rnn}$ is diagonalized such that

$$W_{rnn}^t = C\Lambda^t C^{-1} = C diag(\lambda_1, \lambda_2, \lambda_3...)^t C^{-1},$$

where $C$ and $\Lambda$ are complex. We can then write

$$h_t = C\Lambda_{rnn}^t C^{-1} h_0 + \sum_{i=0}^{t-1} C\Lambda_{rnn}^i C^{-1} W_{in} x_{t-i},$$

which can be written as,

$$C^{-1} h_t = \Lambda_{rnn}^t C^{-1} h_0 + \sum_{i=0}^{t-1} \Lambda_{rnn}^i C^{-1} W_{in} x_{t-i}.$$

To simplify this even more, we can subsume $C^{-1}$ into $\bar{h}_t = C^{-1} h_t$ and $\bar{W}_{in} = C^{-1} W_{in}$, making the activity and input weights complex. We do not calculate $C^{-1}$ directly; instead we treat $\bar{W}_{in}$ and $\bar{h}_0$ as complex and trainable parameters. The equation is no longer equivalent to a real-valued RNN, but it is more general, and we can choose to consider only the real part of the resulting activities. The final equation is

$$\bar{h}_t = \Lambda_{rnn}^t \bar{h}_0 + \sum_{i=0}^{t-1} \Lambda_{rnn}^i \bar{W}_{in} x_{t-i},$$

where $\bar{h}$, $\Lambda_{rnn}$ and $\bar{W}_{in}$ are all complex.

If the modulus eigenvalues of $\Lambda_{rnn}$ are above one, the associated activity rapidly increases, and if they are below one the activity will rapidly decrease. For stability, the modulus eigenvalues must remain close to unity. In order to enforce the condition that $|\lambda_i|$ does not go above one and remains close to one, each complex component of the network can be constructed such that

$$\lambda_i = exp(-exp(\alpha_i) + iexp(\theta_i)).$$

This is the stable exponential parameterization method, as described in Orvieto et al. (2023). The trainable parameters for the network are the set of $\alpha_i$ and $\theta_i$ variables. The output sequence is given by the real part of the hidden states. The network can then be trained efficiently in the parallel form on a GPU using back-propagation. By constructing our network in this parallel form, along with careful parameterization and initialization, we can alleviate the vanishing and exploding gradient problem first noted in Bengio et al. (1994). Ordinarily, the lack of vanishing and exploding gradients is one of the main advantages that transformers have over RNNs.

Now that we have constructed the parallelized LRNN, we can chain them together in blocks and apply multi-layer perceptrons (MLPs) with non-linear activation functions between the blocks. Since we have lost some complexity in the network dynamics of the RNN by choosing a linear activation function, the hope is that by chaining together the LRNN blocks in this manner, we can recover some of the fitting ability of non-linear RNNs. This aspect of the architecture is also described in Orvieto et al. (2023).

## 2.2 Feature Sequence Twisting

We introduce the novel Feature Sequence Twisting (FST) technique to the block LRNN architecture. This is done by transposing the feature and sequence dimensions between each LRNN block. Ordinarily, with an RNN we ask, "How do the elements of a sequence relate given a set of features?" The RNN produces a new representation of the features based on these relationships. When we transpose the feature and sequence dimensions we now ask, "how do the features of a sequence relate given the elements of the sequence?". A new representation of the sequence is then generated based on the relationship between the features. As a loose biological analogy, FST could be thought of as a way of processing a sequence held in long-term memory. By chaining the blocks together and

applying FST the goal is to produce deeper-and-deeper representations of the sequence as well as the features. We present a diagram of the FST architecture in Figure 1. Each FST module contains two LRNNs and two MLPs. The 'hidden feature size' is a hyperparameter of the model and is equal to the hidden state of LRNN 1 and the hidden layers of MLP 1. The second, transposed LRNN 2 and MLP 2 have a hidden state and hidden layers equal to the original sequence length. The coefficients $\alpha_1$ and $\alpha_2$ are defined as $\alpha_1 = \sigma(p_1)$ and $\alpha_2 = \sigma(p_2)$, where $p_1$ and $p_2$ are trainable parameters initialized at zero, and $\sigma$ represents the sigmoid function.

While investigating prior work, we find that the idea behind FST shares some similarities to the "MLP-mixer" architecture from Tolstikhin et al. (2021), which similarly used reshapes and transpositions of the "patches" and "channel" dimensions for computer vision tasks. However, this work relied solely on MLPs rather than blocks of RNNs.

One could apply a similar technique to blocks of transformer networks, transposing the feature and sequence dimensions between blocks. However, because computational costs scale quadratically with sequence length in the self-attention mechanism, this approach would also cause the model to scale quadratically with the hidden feature size. Using LRNNs, we avoid this problem by utilizing an algorithm for parallel computation that scales sub-quadratically.

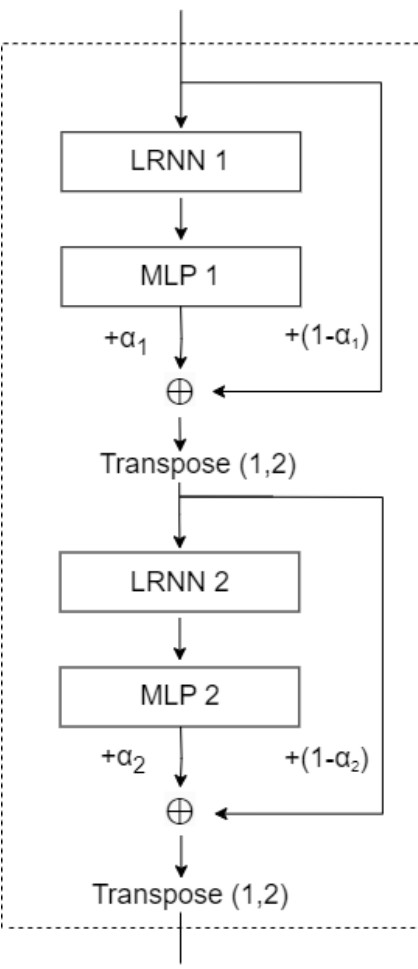

Figure 1: Diagram showing an FST layer with two LRNNs. The multilayer perceptrons (MLPs) use a single hidden layer with a ReLU activation function. The coefficients $\alpha_1$ and $\alpha_2$, which are bounded between 0 and 1, control the proportion of the new representation contributed by the LRNN and MLPs versus the skip connection. Feature-sequence twisting is a reference to the two transpose operations where initial dimensions 1 and 2 refer to the sequence and feature dimensions.

To compute the LRNN in parallel, we used an algorithm presented in Algorithm 1, which was introduced in Yue et al. (2024). This was found to be much more memory efficient than pre-computing all of the powers of the recurrent matrix. This algorithm requires a loop over $\log_2 L$, where $L$ is the sequence length.

---

**Algorithm 1** Algorithm for computing LRNNs, taken from Yue et al. (2024). Here, $W_{in}$ is the input matrix, $\Lambda$ is the diagonal recurrent weight matrix, and $x$ is the input vector.

---

**Input:** $x, W_{in}, \Lambda$
**1. sequence padding**
$L_{\log2} = \text{int}(\lceil \log_2(\text{shape}(x)[1]) \rceil)$
$x = F.\text{pad}(x, (0, 0, 2^{L_{\log2} - \text{shape}(x)[1]}, 0, 0, 0))$
$N, L, D = \text{shape}(x)$
**2. recursive split**
$h = \text{torch.matmul}(x, W_{in})$
**for** $i = 1$ **to** $L_{\log2} + 1$ **do**
   $l = 2^i$
   $h = \text{reshape}(N \times L/l, l, D)$
   **3. parallel forward pass**
   $h1, h2 = h[:, : l//2], h[:, l//2 :]$
   **if** $i > 1$ **then**
      $\Lambda = \text{torch.cat}((\Lambda, \Lambda \times \Lambda[-1]), 0)$
   **end if**
   $h2 = h2 + \Lambda \times h1[:, -1 :]$
   $h = \text{torch.cat}([h1, h2], 1)$
**end for**
**Return:** $h$

---

## 3    RESULTS

### 3.1    LONG LISTOPS TASK

The benchmark task ListOps was introduced by Nangia & Bowman (2018) to evaluate performance of models on sequence-based tasks. It was expanded in the Long Range Arena from Tay et al. (2020) to assess how effective transformers were at computation across a long context length. We present the results of the FST architecture on the Long ListOps task in Table 1. We find that FST with an accuracy of 63.11% outperforms most models, including large transformer models. The exception is the MEGA model, which achieved a slightly greater score of 63.14%, from Ma et al. (2023).

ListOps involves evaluating a nested mathematical operation. For example,

$$\text{Max}(2, \text{Min}(3, 9), \text{Sum Mod}(1, 1), \text{Median}(0, 1, 2), 0),$$

for which the answer would be 3. There are four operators used in Long ListOps; Min, Max, Median and Sum Mod. Sum Mod is the sum of the list, constrained to 0-9 by applying the modulo-10 operator to the result.

The maximum sequence length for Long ListOps is 2000, with an input feature size of 16 tokens and an output prediction range of 0-9. The architecture consists of six sequential FST blocks, each with a hidden feature dimensions of 256. We used an Adam optimizer with a fixed learning rate of $10^{-4}$, a weight decay of $10^{-5}$ and a training batch size of 16. We generated $6 \times 10^6$ training examples using code from the Long Range Arena GitHub repository, and used the default testing set for evaluation, from Tay et al. (2020). This experiment was run on a single NVIDIA RTX 3090 GPU.

There is quite a large gap between transformer models and SSMs on performance in ListOps. It should be noted that the results provided here are for models trained from scratch. It is argued in Amos et al. (2024) that this gap can be closed somewhat by 'self-pretraining' transformers. However, even with self-pretraining these models do not outperform SSMs like S4 (Gu et al., 2021).

Table 1: Comparison of models on Long ListOps. The first set of results are taken from the current github leaderboard of Tay et al. (2020). The S4 result is taken from Gu et al. (2021). The MEGA results are taken from Ma et al. (2023).

| Models | Long ListOps % Accuracy |
|---|---|
| Local Att | 15.82 |
| Linear Trans. | 16.13 |
| Reformer | 37.27 |
| Sparse Trans. | 17.07 |
| Sinkhorn Trans. | 33.67 |
| Linformer | 35.70 |
| Performer | 18.01 |
| Synthesizer | 36.99 |
| Longformer | 35.63 |
| Transformer | 36.37 |
| BigBird | 36.05 |
| MEGA | 63.14 |
| MEGA-chunk | 58.76 |
| S4 | 59.60 |
| FST | 63.11 |

## 3.2 PIXEL-BY-PIXEL MNIST

To demonstrate the limitations of transformers for tasks where information is largely contained within the sequence order, we evaluate the results on a simple sequential MNIST task, where the pixels of MNIST are presented one at a time. This task was first introduced in Le et al. (2015) as a benchmark for sequential data modeling. We used an unmodified transformer encoder with a model size of 512, two encoder blocks, and trainable positional encoding. For comparison, we used two FST blocks with a hidden feature size of 128. Each model was trained with an Adam optimizer with a fixed learning rate of $10^{-4}$. In each case, the target prediction is the output of the final element of the generated sequence. These experiments were run on a single GTX 1080 Ti.

We find that, in half an hour of training, the transformer model is unable to achieve much beyond chance performance. This makes some intuitive sense since the model is fully reliant on positional encoding to transmit information about the sequence order. By contrast, when using an FST model an accuracy of 98.3% is reached quickly.

Since we aim to demonstrate a limitation of the transformer architecture, we do not include convolutional layers or average pooling in this example. This is why this result is different from other studies performed on pixel-by-pixel MNIST using transformers, such as Trinh et al. (2018).

## 3.3 WIKITEXT-103 TASK

In the paper introducing the MAMBA model from Gu & Dao (2024), it is argued that LRNNs and SSMs are incapable of selectively deciding which inputs affect the hidden state passed along the sequence. They argue that it is this property which makes linear time-invariant models such as LRNNs poorly suited to language modeling. However, with FST, we suggest that this argument no longer applies since the representation of the sequence can be selectively altered. To test the language modeling abilities of FST, we use the Wikitext-103 benchmark from Merity et al. (2016). Wikitext-103 contains over 100M tokens of text extracted from Wikipedia articles. The data is split into sequences of token length 512. The perplexity was evaluated using $e^{\mathcal{L}}$, where $\mathcal{L}$ is the cross entropy loss on predicting the next token. We use the 'bert-base-uncased' tokenizer from Devlin et al. (2019), with a vocab size of 30522. While training and evaluating, we use a random number between 0 and 512 to truncate the initial tokens in the data. The model is initialized with six FST blocks and a hidden feature dimension size of 512. We used an Adam optimizer with a fixed learning rate of $10^{-4}$ and a training batch size of 128. These experiments were run on a single NVIDIA A100 GPU.

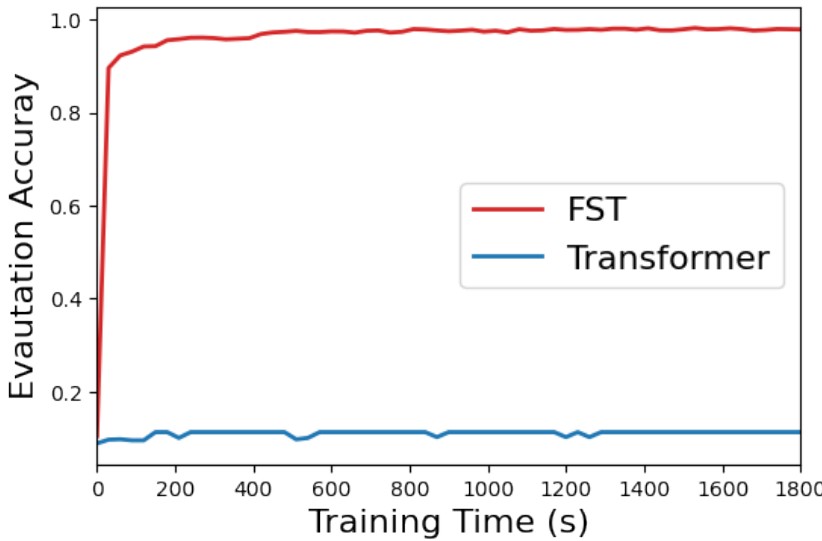

Figure 2: Training an FST model and a transformer model for half an hour on the MNIST dataset where the pixels are presented in a one-by-one sequence. The task demonstrates how difficult it is for the self-attention mechanism to solve tasks where much of the information is contained in the sequence order.

After 770 training epochs, we find a minimum perplexity of 52.83 for the FST model. The state of the art performance using a pre-trained transformer is much lower, for example Hybrid H3 from Fu et al. (2023) achieves a perplexity of 10.6, though the use of pre-trained transformers makes comparisons with a model trained from scratch like FST difficult to interpret. Trained from scratch, the Transformer-XL model from Dai et al. (2019) achieves a perplexity of 18.3. This model uses many more trained parameters (257M) compared to our example (44M).

To make a more appropriate comparison to our relatively small model, we also trained a transformer encoder model with 50M parameters in the same way as we did with our FST example. We use the same tokenizer, fixed learning rate and batch size. This model used trainable positional encoding, a model size of 512, 6 encoder blocks and 8 attention heads. After 659 training epochs we achieved a minimum Wikitext-103 perplexity of 148.93.

### 3.4 SEQUENCE REPRESENTATION AND DEPTH

To see how much the sequence representation is modified in subsequent FST blocks, we can look at the trainable $\alpha_2$ parameters in the model. As shown in Figure 3, these $\alpha_2$ coefficients indicate that the model actively modifies the sequence representation. For the models trained on the Long ListOps and Wikitext-103 tasks,we observe that the $\alpha_2$ coefficients tend to be higher in deeper FST blocks compared to shallower ones. This suggests that deeper blocks rely more heavily on the output from the LRNN to enhance the sequence representation. While this trend is notable, drawing definitive conclusions would require confirmation by training the model on a broader range of tasks.

### 3.5 ABLATION STUDIES

We compared performance on CIFAR-100 from Krizhevsky (2009), presented pixel-by-pixel, using the LRNN model with and without feature-sequence twisting. The sequence length for this task is 1024, with the three color channels as the input features. No convolutional layers are used for the task.

We perform a parameter scan using FST blocks and, for comparison, blocks without the transposed section from Figure 1. We refer to the models without the transposed section as 'LRNN-only' blocks. The comparisons are presented in Figure 4. We also applied transformers with positional

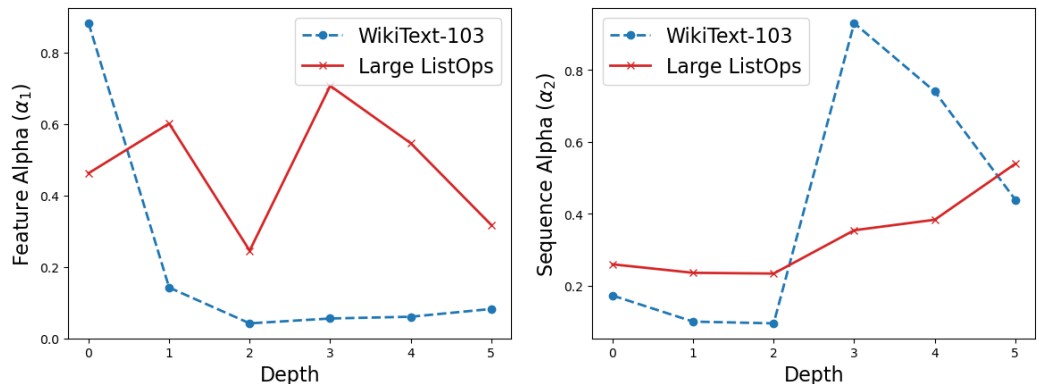

Figure 3: Coefficients for the modification of the features $\alpha_1$ (left) and sequence $\alpha_2$ (right), for different layer depths. Shown for the model trained on Wikitext-103 (dashed blue) and large ListOps (solid red). Presented to show that the model does choose to modify the sequence and that the model also has a tendency to result in a higher coefficient for deeper FST blocks.

encoding, and, similarly to the sequential MNIST task, we find that the model struggles to achieve above chance performance. We use a range of hidden feature sizes [1024, 512, 256, 128] and number of blocks [2,4] for the LRNN-only experiments. For FST experiments we use the same range of hidden feature sizes over [1,2] FST layers. The models were trained using an Adam optimizer with a fixed learning rate of $10^{-4}$, a training batch size of 16 for a total of 40k updates. These experiments were each run on a single NVIDIA GTX 1080Ti GPU.

We find that FST models outperform LRNN-only models at similar parameter sizes. The maximum accuracy achieved by LRNN-only models was 29.95%, while FST models reached a higher maximum accuracy of 32.66%. For FST models, the peak performance was obtained with a hidden feature size of 1024 and 2 FST layers. The LRNN-only models attained their maximum accuracy with a hidden feature size of 512 and 2 blocks.

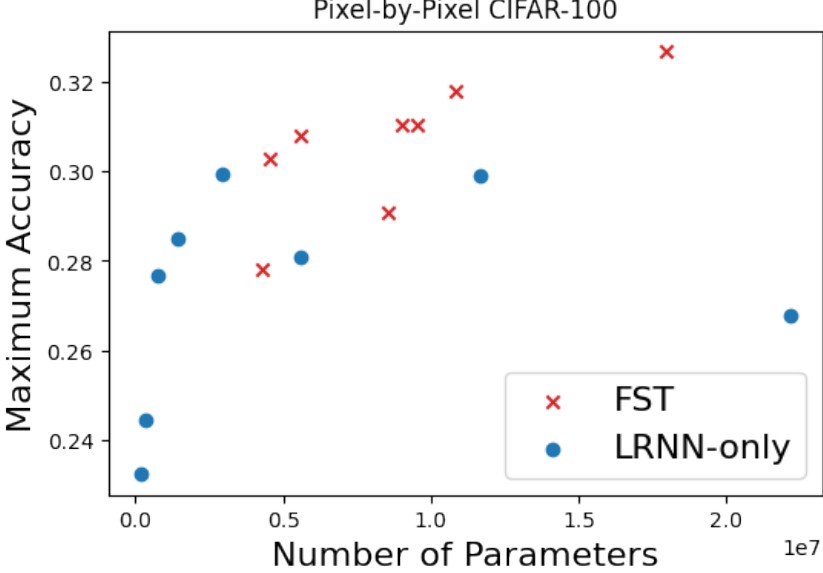

Figure 4: Maximum accuracy on CIFAR-100 presented as a pixel-by-pixel sequence. No convolutional layers. The highest performance is achieved by the FST architecture.

## 4    CONCLUSION

We have demonstrated some advantages of using blocks of FST over transformer models. The FST architecture achieves good performance on the Long ListOps task, competitive with the state of the art. We also show the model's potential for language modeling using the WikiText-103 dataset.

One limitation of the FST model compared to transformers is the lack of an equivalent method to 'attention masking'. It should be noted that LRNNs do not require attention masking for future tokens, as the temporal ordering of elements in the sequence is naturally preserved. However, when the feature and sequence dimensions are transposed after an LRNN block for FST, this property is broken. We can mask certain weights in the FST architecture to ensure that the model is causal. As a future direction for this work, we aim to use this causal version of FST to train with an encoder-decoder structure similar to the way transformers are trained on language translation tasks, for example.

By transposing the feature and sequence dimensions, we also lose a property of RNNs whereby inference can be run on shorter or potentially unlimited sequence lengths. For FST, the sequence length is fixed and will need to be truncated or padded to match the sequence length used during training.

Another limitation is that if we require the hidden feature dimension to be much larger than the sequence length, the cost of computing the LRNN in the transposed mode is expensive. For many practical purposes, however, we find that this is often not the case.

Despite these limitations, we expect that there are many scenarios where using FST is advantageous, requiring fewer computational resources to achieve better results. Overall, we hope that FST will be considered as a competitive and flexible model for use in any sequence-based task. Though this paper has only considered a limited number of benchmark tasks, there is significant potential for the application of FST in more complex tasks. Future work could explore the scalability of FST for longer sequences, including new techniques to condense the information contained within these sequences. We encourage further exploration with FST models to realize its potential across various domains.

## 5    REPRODUCIBILITY STATEMENT

For the purpose of reproducibility, code used for this project is provided as a zip file in supplementary materials. After review, code will be provided on a GitHub repository. We encourage readers to use and build upon the code provided.

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
