# OpenReview forum: "Linear Recurrent Neural Networks with a Feature-Sequence Twist"
_ICLR.cc/2025/Conference — ICLR 2025 Conference Withdrawn Submission_

### Official Review · Reviewer_9cWg · 2024-10-31

**Soundness:** 1
**Presentation:** 3
**Contribution:** 1
**Rating:** 3
**Confidence:** 4

**Summary:**

This paper introduces a feature sequence twisting module that combines a sequence processing layer and a feature processing layer using linear RNNs.
Experiments on LonglistOps, MNIST and wikitext indicate favorable performance compared to various transformer-like models.

**Strengths:**

Overall, the paper is clearly written.

**Weaknesses:**

- (novelty) The idea of alternating sequence and feature processing/mixing has been introduced by (Tolstikhin et al., 2021).
   The argument that MLP-mixer uses MLPs instead of linear RNNs is not sufficient to claim novelty in this regard.
   Also, the linear RNNs can be readily taken from (Orvieto et al., 2023).
 - (significance) By requiring a fixed sequence length, the Linear RNN effectively turns into an MLP, albeit with a different parameterisation.
   This connection, which would make the proposed model equivalent to MLP-mixer, is not discussed in the current manuscript.
 - (significance) The proposed module is only proposed for LRNNs, but there is no motivation for why.
   The module seems to be general enough to be applied to a variety of models, but this is not considered in the current manuscript.
 - (quality) Performance of LSTMs on ListOps reaches over 74% accuracy (Nangia & Bowman, 2018).
   This seems to indicate that traditional recurrent networks should be able to perform well.
   However, these have not been considered in this baseline and it is unclear whether these models provide any benefits over recurrent networks.
 - (quality) There are no error bars to indicate the consistency of the different models.
   Especially for the close results on ListOps, it would have been useful.
 - (quality/clarity) The MNIST experiments hints at tranformers missing positional encodings.
   Using transformer models without positional encoding for sequence modelling makes little sense and leads to an unfair comparison.
 - (quality) Instead of scaling up the FST model, the transformer baseline was scaled down for the comparison on Wikitext-103.
   Apart from the fact that this seems to indicate the model does not scale,
   there are little to no details on how the hyper-parameters were tuned for the baseline.

**Questions:**

1. What are the connections and differences between this work and MLP-mixer?
 2. Is it possible to use the LRNN parameterisation to build an MLP?
 3. Why is the FST layer introduced only for LRNNs?
 4. How does this model compare against an MLP-mixer?
 5. How do traditional recurrent models compare to the proposed LRNN?
 6. How consistent is the performance of the model?
 7. What positional embeddings were used for the transformer baselines?
 8. Does the model scale?

---

### Official Review · Reviewer_VyVD · 2024-11-01

**Soundness:** 2
**Presentation:** 2
**Contribution:** 1
**Rating:** 3
**Confidence:** 4

**Summary:**

In this paper, the authors revisit traditional linear recurrent neural networks (LRNN), introducing Feature-Sequence Twisting (FST), a novel structure that transposes the sequence and feature dimensions between LRNN blocks. The purpose of FST is to generate deeper representations of the sequence in subsequent LRNN blocks. Moreover, FST remains practical to compute even for large feature dimensions.

**Strengths:**

-

**Weaknesses:**

1.The biggest problem with this paper is the insufficient contribution and poor expression. The authors propose a transposed LRNN, which is merely a simple variation of the MLP-mixer. The authors also fail to provide adequate experimental evidence to demonstrate the advantages of this structure.

2.The authors claim that the purpose of this paper is to generate deep representations of sequences. However, the experiments do not reflect this claim: the tasks related to sequences are overly simplistic. The authors could refer to how other papers conduct sequence experiments, such as TimesNet [1]. Both image tasks lack extensive comparisons, and in Figure 2, the accuracy of the Transformer is artificially lowered to 20%, which is impossible (ViT is the best in image classification). The paper records a Cifar100 result with only about 30% accuracy, indicating severe underfitting, which does not allow for an accurate assessment of the strengths and weaknesses of the two models.

[1] TimesNet: Temporal 2D-Variation Modeling for General Time Series Analysis

3.The paper's expression is very poor, with many incorrect uses of quotation marks; most references lack correct sources.

**Questions:**

See weekness.

---

### Official Review · Reviewer_1YFr · 2024-11-03

**Soundness:** 2
**Presentation:** 1
**Contribution:** 1
**Rating:** 1
**Confidence:** 4

**Summary:**

This paper introduces Feature-Sequence Twisting (FST) as a variation of linear recurrent neural networks (LRNNs) with transpositions in the feature and sequence dimensions. The approach aims to improve sequence representations in LRNNs, with the idea of enhancing sequence-based tasks by allowing bidirectional processing. Experiments demonstrate modest improvements on a few benchmark tasks compared to baseline models.

**Strengths:**

* Clear explanation of the FST concept

**Weaknesses:**

* Not novel: This is a bidirectional version of a previous paper with no methodological improvements other than making it bidirectional.
* Limited emperical evidence

**Questions:**

The introduction has a page of RNN/transformer history. This is unnecessary. Just cut to the point.

Section 2.1 spends an entire page highligthing another paper's results. This is also unnecessary.

---

### Official Review · Reviewer_nGbA · 2024-11-05

**Soundness:** 2
**Presentation:** 1
**Contribution:** 2
**Rating:** 3
**Confidence:** 4

**Summary:**

This paper introduces feature sequence twists and couples them with LRNNs to efficiently interact between various modules, which could potentially help LRNNs perform comparable or better than transformers on various NLP downstream tasks. Authors have conducted small-scale experiments on various datasets to show the advantage of their proposed approach.

**Strengths:**

1. Finding alternatives to transformers is important, and LRNN and S4 are promising approaches.

**Weaknesses:**

1. The paper seems to be rushed; the writing needs improvement.
2. Incremental work -- MLP mixer is similar; the only update is adding LRNN, which is again added to stateless MLP. Hence nothing novel
3. Ablation Study Missing
4. Large-scale experiments missing
5. Entire Related Work Section is Missing -- Echo state networks, Reservoir computing, newly proposed xLSTM, various SSM models, RWKV (combines RNNs with Transformers).
6. Sequential Overfitting

**Questions:**

Few Additional Questions

In the manuscript, authors say that “The ’hidden feature size‘ is a hyperparameter of the model and is equal to the hidden state of LRNN 1 and the hidden layers of MLP 1” –isn’t this computationally expensive if the state dimension of LRNN1 is 256, MLP1 would require those many layers. Or did the authors mean the hidden dimension of MLP? If yes, then how many layers does MLP have? None of this information is mentioned in the manuscript.

Next, the authors mention, “The second, transposed LRNN 2 and MLP 2 have a hidden state and hidden layers equal to the original sequence length.” – same query: does the author mean hidden layer dimensions? How many layers does MLP have in this case? Another issue is computational cost. If the sequence length is 10k, that would mean LRNN2 and MLP2 both have 10k hidden dimensions. The authors don’t mention much about these issues.

In Figure 1, what is the significance of having (1 – sigmoid(p1/p2)), and why is the ‘+’ symbol added after  +(alpha_1), +(alpha_2)? Sigmoid will always result in values between 0-1, so having + is redundant. This brings us to the next question: where are you getting p1 and p2? They are only mentioned on line 167. Is it linear transformation from prior layers or direct output of those layers? This brings us to the next question: given the output size of LRNN2, constant application of sigmoid will lead to vanishing and exploding gradients? So, how can authors claim that the proposed design overcomes vanishing and exploding gradient issues?

Ablation Study: Why only 2 LRNNs? What if you have 1 or 3? Do the results change? What about the number of layers for MLP? Isn’t that also a hyper-parameter? Detailed analysis should be done across these settings.

Experimental Setup: The authors should mention all the hyperparameters used for LRNNs and transformers, not just final settings. There is no mention of hyperparameter optimization. It is well-known that transformers with fewer parameters tend to collapse, and prior works like MEGA have shown better implementation of transformers, which leads to better performance.
The biggest concern is performing sequential overfitting, the models are optimized by seeing performance on validation/evaluation set. Thus, results cannot be trusted. One can ideally get basic_train.tsv, basic_val.tsv, basic_test.tsv from lra data repo, same goes for MNIST and CIFAR.

Finally, there is no statistical significance testing, so it's difficult to truly judge the method's importance. Authors should also report the following stats: the Total number of parameters for both models, Convergence time (the proposed approach reaches faster convergence on all), per epoch training time, and FLOPS.

---

### Note · Authors · 2024-11-13

**Comment:**

Reviewers have highlighted significant concerns with the work. We do not believe we can adequately address these concerns within the rebuttal timeframe; consequently, we have decided to withdraw the paper.

While we are naturally disappointed with this outcome, we thank all reviewers for taking the time to provide valuable comments, which we will carefully consider in future work.

**Withdrawal Confirmation:**

I have read and agree with the venue's withdrawal policy on behalf of myself and my co-authors.